# Multi-Dimensional Comparison of the Impact Mechanism of the Self-Rated Health Status of Urban and Rural Residents in Chinese Social Environments

**DOI:** 10.3390/ijerph191710625

**Published:** 2022-08-25

**Authors:** Chao Yu, Xinyi Zhang, Junbo Gao

**Affiliations:** 1School of Geographic Sciences, Xinyang Normal University, Xinyang 464000, China; 2The Center of Targeted Poverty Alleviation and Rural Revitalization, Xinyang Normal University, Xinyang 464000, China; 3School of Tourism, Xinyang Normal University, Xinyang 464000, China

**Keywords:** multi-dimensional comparison, self-rated health status, urban and rural residents, social environments, China

## Abstract

Self-rated health status (SRHS) reflects individuals’ social environment, and the difference between urban and rural areas in China further highlights the impact of social environment on health. This paper aimed to systematically analyze and compare the impact mechanism of the SRHS of urban and rural residents from multiple dimensions, i.e., time, space, and scale. Drawing on data from the Chinese General Social Survey (CGSS) and China Statistical Yearbook, we used spatial, cross, and HLM analyses. Results indicate that: (1) From 2010 to 2017, the overall SRHS level of Chinese residents gradually declined; the gradient pattern of east, middle, and west became more marked, and the health level in rural areas generally fell behind that of urban areas. (2) The focus of SRHS moved toward mental health, and people’s perceptions of the social environment gradually became a key factor affecting health. (3) In the long term, the gradient allocation of medical service resources could narrow the gap between urban and rural areas to comprehensively improve regional health levels.

## 1. Introduction

Health has always been an important topic for humans because it forms the basis of the normal operation of all social and economic activities. Rapid urbanization has impacted the health of Chinese people, leading to many changes. From a positive perspective, urbanization has brought social and economic prosperity and improved medical and health conditions in many parts of China. However, it has also widened the omni-directional gap between urban and rural development [1]. The impact these social environment differences have on the health of urban and rural citizens has become the focus of research [2].

The concept of “health” used in disciplines such as sociology and geography differs from a medical perspective involving clinical tests and diagnoses. Sociologists and geographers frequently evaluate peoples’ health based on their completion of health-related questionnaires or self-rated health status (SRHS) [3]. SRHS also reflects respondents’ current social environments to some degree. Thus, describing the urban and rural conditions that best reflect the regional differences in China, and comparing the influence mechanisms on urban and rural residents’ SRHS may deepen our understanding of the impact of the social environment on health.

When scholars study the impact mechanism of SRHS, they usually rely on social environment (economic) models. Among them, income, education, and employment are the three most commonly used indicators included in the model [4]. Despite differences in methods and classification of indicators, many researchers have found that health conditions, such as morbidity or self-rated health, vary stepwise with the social environment (economics). However, there are still many inconsistencies in the mechanisms of each factor. For example, income includes many types such as absolute and relative, personal and family. The negative effects of relative income on health manifestations are often underappreciated, and income inequality may trigger “income-healthy poverty” traps [5]. Adequacy of disposable income may be a better indicator than absolute income, particularly for older women [6]. Family income was strongly associated with both SRHS and the presence of a longstanding illness or health problem, and there were also significant gender differences [7]. Education usually refers to the highest educational level achieved. Higher education is associated with better health [8], and lower levels of SRHS occur mainly among middle-educated or single adults [9]. Employment focuses on the employer-employee relationship and job characteristics such as work experience and hours. A phenomenon called the poor worker’s long working hours paradox refers to how the health effects of long working hours appear to be diminished in low-income households [10]. Similarly, informal employees or self-employed individuals were more likely to report good health than regular employees because of reduced conflict between work and family [11]. In China, the overall health level of state workers is higher than that of other groups [12].

The complex influence of income, education, and employment on SRHS indicates that there are many factors affecting health in a range of ways. Scholars have approached this complexity through in-depth research in different fields, which can be roughly divided into two areas. One area focuses on individuals and families, examining factors such as their living conditions, health literacy, and medical insurance. The main conclusions from this approach often stand in contrast to common sense. For example, people with good living conditions may have poor SRHS because of high health expectations or stress about paying their mortgages [13]. Personal attributes such as gender, age, education, and occupation are related to health, but health literacy is the key determinant of health [14]. Students’ SRHS was not improved by enrollment in a unified school-based private health insurance plan although individualized health insurance significantly improved students’ SRHS [15]. Media promotion had a positive effect on purchasing private medical insurance, whereas personal awareness had a negative effect [16]. The second approach involves exploring the impact of ‘place’ on health from a broader perspective than the individual (e.g., focusing on the community, city, or province). The role of place in health began to gain traction at the end of the 20th century [17]. Now, for example, links between green space and health are widely acknowledged [18]. Green space has a positive effect on improving SRHS [19], and urban residents are more aware of the benefits of green areas than rural residents [20]. However, the impact of the social environment on health is mediated through environmental perception rather than actual conditions [21]. People’s perceptions of the living environment and neighborhood problems had an impact on all aspects of well-being, suggesting that research on the impact mechanism of SRHS should incorporate higher-level objective conditions of the social environment [22], along with people’s subjective feelings about regional comparisons.

In terms of urban and rural disparities in China, studies have shown that urban and rural residents exhibit non-linear differences in SRHS and that SRHS has a slight urban advantage [2]. Medical insurance and green space are two examples of the most common differences between urban and rural areas. Urban medical insurance is more significantly related to the health of migrant workers than rural medical insurance [23]. The association between residential green space and health is stronger in high-density urban areas than in low-density urban and rural areas [24]. As a result, rural residents have higher rates of chronic disease and lower self-rated life expectancy [25]. However, rural residents also have some advantages. Residents migrating from rural to urban areas generally face social integration issues, and the indirect impact of social integration on mental health needs to be mitigated by weakening the buffering effect of social integration on perceived stress [26]. Greater emotional stability in older rural adults also plays a key role in buffering the negative correlation between health and well-being [27].

In summary, the fairly large literature on social environment and health shows that the impact mechanism of SRHS follows certain rules in time trends, urban and rural spatial pattern, and individual-living environment nesting level. However, research lacks understanding of the integration of these multiple dimensions of time, space, and scale. Considering time and space—but lacking scale—makes it hard to explore the relationship between individual health and living environment. When time and scale are considered—but not space—the differences in health status between regions cannot be explained. Considering space and scale only, it is impossible to conclude the changing trend of influencing factors. SRHS is a result of the comprehensive effect of people’s perception of the social environment, and study from a single dimension may overemphasize the influence of a certain factor. To address this challenge, this paper attempts to systematically analyze the impact mechanism of SRHS by integrating multiple dimensions of time (2010–2017), space (urban and rural areas), and scale (individual–provincial region) with Chinese residents. We set out to answer the following questions: (1) What are the characteristics of the spatial pattern of SRHS? Is it consistent between urban and rural areas? Is it consistent with China’s social and economic development pattern? (2) What is the impact of individual and provincial levels on SRHS? Which level of difference between urban and rural areas is greater? (3) How has the spatial pattern changed over time? What is the trend of the influence of different factors? Through in-depth investigation of the key factors influencing health, this paper aims to provide a targeted reference when formulating health optimization strategies for individuals at different levels.

## 2. Materials and Methods

### 2.1. Data Source and Processing

#### 2.1.1. Individual Data

Data source

The data used in this paper came from the Chinese General Social Survey (CGSS) in 2010, 2013, and 2017. CGSS was China’s first comprehensive, nationwide coherent academic survey program, in place from 2003. Following international standards, CGSS adopts a multi-level stratified random sampling plan to collect data comprehensively and systematically from multiple levels—individuals, families, communities, and provinces—to effectively represent Chinese society. The privacy protection principle of CGSS means that community level information is hidden and a nested data structure could not be formed. Thus, the community level was not incorporated in the model construction.

2.Data processing(1)**Dependent variable.** The dependent variable came from the survey of SRHS in CGSS, using a five-point scale. To ensure concise and comparable data, we converted the scale into a binary format, namely “healthy” (indicated by 1) and “unhealthy” (indicated by 0) (Table 1). “Very healthy”, “relatively healthy”, and “average” on the original scale were classified as “healthy”, and “unhealthy” and “very unhealthy” were classified as “unhealthy”.(2)**Independent variable.** At the individual level, we included information such as location, personal and family attributes, income level, work status, social security, physical exercise, happiness, and information sources (Table 1). First, location is a sign of the connection between levels of the multi-layer model that includes both urban and rural differences and inter-provincial differences. Second, at the individual level, we added two indicators—happiness and information sources—in addition to education, marriage, income, work, insurance, and exercise, used in previous studies. Happiness is a subjective feeling, such as SRHS, but a close relationship between the two constructs has been agreed by scholars [28]. The information source indicator was mainly used to reflect the ubiquitous role of the Internet in ensuring that people can conveniently access knowledge and consultations about their health. This access to information ultimately serves to improve daily living habits and detect changes in individuals’ health status as early as possible [29]. Finally, as for the dependent variable, we simplified some of the measurement options for some of the independent variables. This was done in two ways: by transforming responses into binary data, and reducing categorical items such as educational level. Where key information in the questionnaire was missing, these data were deleted. After removing incomplete data, a total of 27,894 sample data remained. Further inspection of data distribution showed that the distribution of sample data across provinces was consistent for the different years, with a mean percentage of 3.57% for each province in each year. Therefore, the missing values had no significant effect on the study.

#### 2.1.2. Provincial Data

Data source

The provincial level covers 28 of the 34 provincial-level regions in China’s mainland (excluding Hainan, Tibet, Xinjiang, Hong Kong, Macao, and Taiwan). Most of the survey content in CGSS is personal and family information was gathered 6 months to 1 year previously. Thus, the provincial data came from the China Statistical Yearbook in 2010, 2013, and 2017, corresponding to the data from the end of 2009, 2012, and 2016, and including both urban and rural information. For consistency, subsequent analysis results are identified as 2010, 2013, and 2017.

2.Data processing

To reflect differences in the social environment, provincial-level data include the level of public health service, burden of health care and medical service expenditure, population structure, and the convenience of public facilities (Table 2). The level of public health service and the burden of health care are directly related to residents’ health, and there are significant differences between urban and rural areas among provinces. Population structure is reflected by the degree of aging and the level of education. The aging population in China is increasing, and the elderly have a high demand for health and medical services [30]. Improved education levels may deepen understanding of health and increase individuals’ willingness to seek medical treatment, but it may also negatively impact health status because highly educated people are more likely to experience work pressure [31]. Therefore, young and elderly people with higher education may face greater risks to their health status, ultimately affecting the overall level of health and medical services in the region. For public facilities, the focus is on factors that may affect health, such as water use, park green space, and garbage disposal. Changes in statistical quality meant that alternative indicators were selected for some years. Among them, PF_WA, PF_GN, and PF_GA were used for urban data each year but only for rural data in 2017. Instead, PF_PWA, PF_ITO, and PF_STO were used in 2010 and 2013 for rural data.

### 2.2. Methods

Scale change is an important approach in geographic research, reflecting the multi-layered structure of research questions. Health is related to personal physical conditions and family living habits, and the social environment in which people live. China’s population household registration policy ensures that all citizens are embedded in the administrative governance of “province (municipality, autonomous region)–city (state)–county (district)–township (street)–village (community)–individual (family)”. Therefore, analyzing the impact mechanism of SRHS from all these levels is more in line with reality. The basic prerequisites of traditional linear regression analysis methods are linearity, normality, homogeneity of variance, and independent distribution. However, homogeneity of variance and independent distribution do not hold for multi-level nested data. Thus, we used a hierarchical linear model (HLM), widely used in environmental and health research [32]. This paper involves two levels of different individuals and provincial regions. The two-level model can be simplified as:(1)$$y_{ij}=\beta_{0}+\beta_{1}x_{1ij}+\beta_{2}x_{2ij}+\beta_{3}x_{3j}+\beta_{4}x_{4j}+\varepsilon_{ij}+\mu_{j}$$
(2)$$\mu_{i}∼N\left(0,\sigma_{\mu }^{2}\right)$$
(3)$$\varepsilon_{ij}∼N\left(0,\sigma_{\varepsilon }^{2}\right)$$

In the formula, *i* and *j* represent different individuals and provincial regions, respectively. *β* is the regression coefficient, *x* is the independent variable, and *ε* and *μ* are the residual items at the individual and provincial levels, respectively. Considering that the dependent variable used in this paper comprises binary data, we needed to perform ‘Logit Transformation’ on the dependent variable [33].

Data analysis took place using R, a free software environment for statistical computing and graphics, with a range of packages for researchers. In this research, we used the package ‘bruceR’ (version 0.8.5) by Han-Wu-Shuang Bao (Beijing, China) for HLM analysis based on R (version 4.1.3).

## 3. Results

### 3.1. Spatial Analysis

(1)**Chinese residents’ levels of SRHS had declined**. From 2010 to 2017, the sample quantity increased, but the structure remained the same. In 2013, the ratio of rural to urban data was 1.5:8.5, and for the other two years, the ratio was 2:8 (rural to urban). The change in standard deviation shows that the SRHS difference among Chinese residents expanded, especially for rural residents whose annual standard deviation was larger than the full sample level. The proportion of healthy groups of Chinese residents declined but remained at a relatively high level (Table 3). The health status of urban residents was relatively stable, whereas the proportion of unhealthy rural residents had increased significantly, providing the main reason for the decline of the overall health level.

(2)**The SRHS showed significant spatial gradient distribution characteristics.** The health status of people in the eastern coastal areas was relatively higher than those in other areas, and the healthy ratio consistently remained above 80% (Figure 1). In the central region, except for Shanxi, the health level was relatively high, but other provinces reported mid-levels of health. The western region had poor health status—the proportion of healthy people in Gansu never exceeded 60%, possibly related to the outdated public services in that region. The health levels of Heilongjiang and Jilin in Northeast China declined each year, which may be related to the serious population outflow and increased aging population in Northeast China. When the sample was divided into urban and rural groups, the spatial gradient of health status became more obvious, especially the health status of rural residents. Urban residents’ health status was generally high, and the areas with lower health levels gradually shrank. Rural residents showed the opposite trend. The proportion of unhealthy groups and regions continue to expand, and the health status of rural residents in the northwest, southwest, and northeast regions was poor.

(3)**There were obvious shortcomings in rural public services.** Further comparison of the social environment of provincial-level regions showed little difference between urban and rural aging and the proportion of health and medical consumption. However, there was a large gap between urban and rural medical staff and the number of hospital beds, health and medical consumption expenditure, and education level (Figure 2). Correlation analysis also showed that the level of medical services, income, medical expenses, and education levels in rural areas were significantly correlated, but the correlations between these factors in urban areas were not significant (Figure 3). In fact, by the end of 2016, the urbanization rate in China was 57.35%, and the urbanization rate in eight provinces did not exceed 50%, including provinces with poor health levels such as Gansu, Sichuan, Yunnan, and Guizhou. Therefore, given the large rural population, rural public services had obvious shortcomings. The strength and changing trend of these specific influences required further testing by HLM analysis.

### 3.2. Cross Analysis

(1)**Married men reported better health.** Although the average life expectancy of Chinese women was significantly higher than that of men (According to the China Statistical Yearbook, the average life expectancy of Chinese men in 2015 was 73.64 years, whereas the average life expectancy of Chinese women was 79.43 years.), the results showed that men’s health was better than women’s in urban and rural areas (Table 4). The gap between the two was not obvious in cities, but more prominent in rural areas. This finding may be related to the fact that there are more women left behind in rural areas who must undertake multiple tasks such as agricultural production, children’s education, and elderly care. As a result, the health rate of unmarried women in rural areas was below 70%, dropping to 66.08% in 2017. Correspondingly, the mutual support of family members was a strong guarantee of health, and the health of the married population was better. Although physical fitness brought health benefits, there was no significant difference linked to whether BMI was above the norm.(2)**Public officials working full-time on non-farm jobs were in better health.** Differences in health status caused by occupational attributes were the most prominent. More than 90% of residents engaged in non-farm work in urban or rural areas were consistently healthy (Table 4). Farm workers face long-term physical demands and part-time work compresses leisure and exercise time. These workers usually have a relatively low income level and poor health. Working in a government department (usually publicly owned) is associated with relatively sound social security and employee welfare systems, and urban public officials generally had better health in terms of job attributes. Higher-level (urban) departments in China typically pay better salaries than lower-level (rural) departments. Examining medical insurance participation showed no significant difference in the impact of medical insurance on SRHS—medical insurance did not negatively affect health, but rather reflects China’s very high participation rate. Survey data showed that over 90% of urban and rural residents had medical insurance, most likely because of China’s targeted poverty alleviation strategy since 2013, with its core policy of basic medical insurance. The wide medical insurance coverage has led to a significant improvement in the medical security level of the rural population.(3)**Happiness was a strong support for physical and mental health.** Of all individual-level attributes, happiness had the most significant impact on SRHS. Even in 2017, the proportion of unhealthy rural residents who considered life unhappy exceeded 50% (Table 4). Regardless of urban or rural residence, the health status of the unhappy population was relatively poor and showed a worsening trend. Happiness and SRHS may be mutually causal. Both are the result of self-assessment, and the physical and economic burdens caused by other factors (e.g., diseases) also affect perception of happiness.(4)**Moderate exercise was good for health.** Although physical exercise can improve immunity and maintain physical function, the intensity of exercise varies from person to person, and more exercise is not always better. The results of the survey showed that people who performed high-intensity physical exercise several times per month had the best SRHS (Table 4); among them, the health rate of rural residents was consistently above 80%, and that of urban residents consistently at 90%.(5)**New media promoted the efficient dissemination of health knowledge.** The ‘2010 Chinese Language Life Status Report’ issued by the Chinese Ministry of Education pointed out that 2010 was the first year of Weibo and the Internet ecology with Weibo as the carrier. With its fast information transmission and strong fidelity, Weibo quickly occupied an important position in people’s lives. In 2013, the launch of the “4G” network and the widespread popularity of smartphones prompted Chinese residents to enter the era of high-speed communication amid the rapid development of mobile terminal software and hardware. Every resident can now use a smartphone to get real-time health information through Weibo and other applications, greatly improving people’s awareness of health management and disease prevention. The survey data strongly showed the important influence of technological innovation. The SRHS of people who used new media as the main source of information reached an excellent level in the initial stages (2010 and 2013), but it declined significantly in 2017 (Table 4). After the explosion of the information scale, the proportion of false information also increased significantly, possibly impacting people’s judgments. The increased dependence on mobile phones has also produced many negative habits that affect people’s health.

### 3.3. HLM Analysis

(1)**The impact of the individual level on the SRHS of urban and rural residents was the same.** The individual level contained 19 variables, and 12 of these showed a significant impact on the SRHS, covering all categories except social security (Table 5). Among them, age, work experience, and happiness had significant effects on urban and rural residents each year; education and family income were consistently important factors for rural residents, whereas exercise frequency was consistently important for urban residents. The remaining variables differed greatly between urban and rural areas across years. The influence of each variable on SRHS was consistent with the previous analysis, that is, people with good health status were mainly young and well-educated, engaged in non-farm work with a high family income, exercising regularly, actively receiving the latest information, and with a strong sense of happiness.(2)**The impact of the provincial level on the SRHS of urban and rural residents varied greatly.** The provincial level contained 14 variables, and five showed a significant impact on SRHS, covering all categories at the provincial level except population structure (Table 5). No variable had a significant impact both in urban and rural areas. Among them, medical technical persons, licensed doctors, and the public green area had a significant impact on rural residents, and medical expenses and their proportions had a significant impact on urban residents. All variables were only significant in 1 year except for the proportion of medical expenses, which always had a negative effect.(3)**The changing trend in the influence of each variable reflected the difference in demand between urban and rural residents.** At the individual level, the influence of education on rural residents gradually increased, reflecting the general improvement in the quality of the rural labor force (Table 5). The influence on urban residents had declined, which may be related to the greater mental pressure faced by groups with higher education. The influence of work experience on rural residents declined, possibly related to the common part-time work pattern (linked to seasonal farming work) of rural residents. The influence of happiness on rural residents also decreased, but its influence on urban residents is increasing—perhaps linked to the pursuit of a high-quality life by urban residents. The influence of BMI and information sources were only highlighted in the first 2 years, indicating that most people already had awareness of body management as a key factor affecting future health. At the provincial level, rural areas had changed from supplementing medical resources in the early stages to constructing public green areas in the later years, and overall health-related infrastructure had improved. The infrastructure in urban areas was relatively sound, but urban residents had long faced the problem of a high proportion of medical expenses. In the process of gradually improving the infrastructure in rural areas, rural residents are also likely to face similar problems. The cost of medical insurance in China has nearly doubled from CNY 180 in 2017 to CNY 350 in 2022—for all residents. For rural residents with an annual per capita income of less than CNY 20,000, that burden is very heavy.

## 4. Discussion

(1)**Narrowing the gap between urban and rural areas is key to improving regional health.** The vast gap between urban and rural areas—whether at the individual or provincial level—and the overall shortcomings of rural areas significantly affect the health level of China and its regions. More concerning, the gap between urban and rural areas has not narrowed significantly [34] and the health of rural residents continues to deteriorate. In the process of improving the health of rural residents by increasing their income, a population migration route from rural areas in undeveloped provinces to cities in undeveloped provinces to cities in developed provinces has gradually formed, making urban-rural differences deeply bound to provincial differences [35]. The fact that the spatial pattern of health rate was consistent with the pattern of China’s social and economic development illustrates this point. The resulting cumulative effect has caused an imbalance in the regional high-quality labor force and also enhanced people’s sensitivity to the social environment, magnifying the influence of the social environment on health in regional comparisons.(2)**Multiple challenges in improving rural health.** We incorporated all the variables in this paper—and their influences—into a unified framework (Figure 4). Urban and rural differences are significant, along with regional differences. This verifies the necessity of analyzing the impact mechanism of SRHS from different levels and scales, such as urban and rural areas and provincial regions. The performance of the variables suggests an underlying reason for the differences in work attributes, living customs, social welfare, and other aspects caused by the functional orientation of urban and rural areas [36]. The nature of their work and living customs means that rural residents, especially farmers engaged in agricultural production, have significantly higher exposure risks to environmental pollution than urban residents [37]. Therefore, rural residents are more sensitive to differences in the social environment and eager to change their working and leisure environments to places comparable with those in cities. However, economic development concentrates in cities and rural areas are faced with the contradiction of strong demand but insufficient investment [38]. Although urban and rural integration is key to achieving rural revitalization, a one-way flow from the countryside to the city is more common. This uni-directional movement may have obvious effects of improved health for individuals but has a minimal impact on promoting regional health. This reminds us that although health is a manifestation of individual conditions, improving health requires transformation of the individual—and more importantly—of the social environment. Therefore, in response to the problems of happiness, green space, and the widespread aging of rural areas that rural residents care about, improvement measures are needed in the following areas. First, we should carry forward traditional culture and make full use of the rural stage. While enhancing the sense of local identity and belonging, this would also support the health of the villagers. Second, with rural “left-behind” women as the main labor force, a new mode of home-based care for the elderly should be explored in rural areas under the unified arrangement and supervision of village committees. The third is to deepen the hierarchical diagnosis and treatment system. This would ensure that common and chronic diseases are treated conveniently and efficiently in the county, ultimately improving rural residents’ willingness to seek medical treatment.

## 5. Conclusions

This study systematically analyzed and compared the impact mechanism of the SRHS of urban and rural residents from multiple dimensions, such as time, space, and scale, based on the data of CGSS and the China Statistical Yearbook. The main conclusions are as follows:(1)The SRHS of Chinese residents showed a temporal trend of gradual decline in overall level, a spatial gradient pattern with gradual deterioration in the east, the middle regions, and the west, and a regional difference where rural regions were generally less developed. Multi-dimensional comparative analysis provided a more comprehensive and in-depth understanding of the impact mechanism of SRHS, and strengthened understanding that narrowing the gap between urban and rural areas in China is key to improving regional health.(2)The performance of variables such as happiness, information sources, and leisure space reflects how SRHS is tilting toward mental health. Medical insurance coverage in China has risen to a high level and the influence of social environmental pressure on health has gradually become dominant. In the future, it is necessary to increase the investment in the mental health industry while continuing to improve infrastructure construction and promote the development of the health industry.(3)Spatial patterns and urban-rural differences indicate that regional health levels are consistent with socioeconomic development stages. Migrating to developed areas with a better social environment can only improve personal health in the short term, and such individuals will still face high costs of urban medical services in the future. Only by further eliminating the differences in the sensitivity of urban and rural residents to the social environment and by grading the allocation of medical service resources can regional health levels be truly improved.

This paper has limitations associated with the data and methods used: (1) Binarizing the dependent variable is convenient for research, but is rather absolute. Current discussions about “sub-health” are gradually increasing [39] and, in the future, we should increase the sub-health gradations between “healthy” and “unhealthy” to analyze the mechanism of the transition from sub-health to healthy or unhealthy. (2) We only discussed health at the provincial level because of the concealment of community-level information in CGSS. Future work should consider adding field research to questionnaire administration to build a model more in line with the administrative governance system. (3) The current research time node was 2017. Although the implementation of the targeted poverty alleviation strategy was included, the COVID-19 pandemic exerted a profound impact on global health. Rural areas, in particular, faced great challenges with their healthcare systems [40]. In the future, the impact of sudden infectious diseases outbreaks or other disasters on regional health will become a new research direction.

## Figures and Tables

**Figure 1 ijerph-19-10625-f001:**
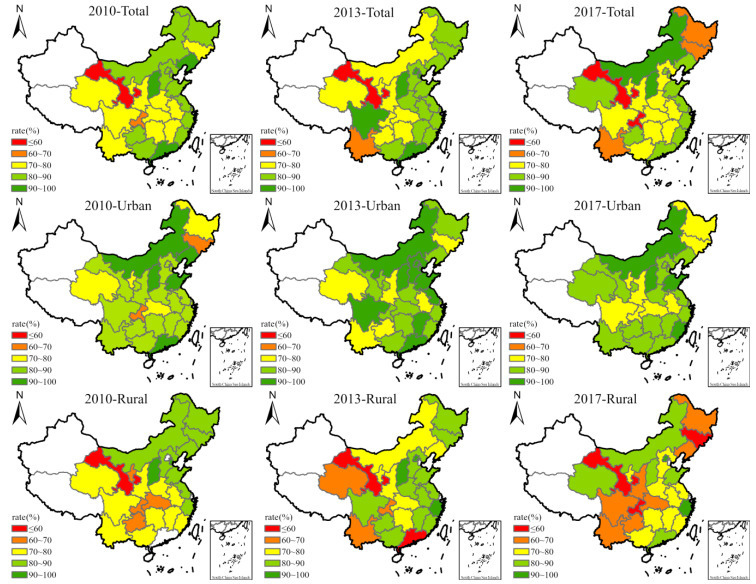
Spatio-temporal patterns of the proportion of healthy people.

**Figure 2 ijerph-19-10625-f002:**
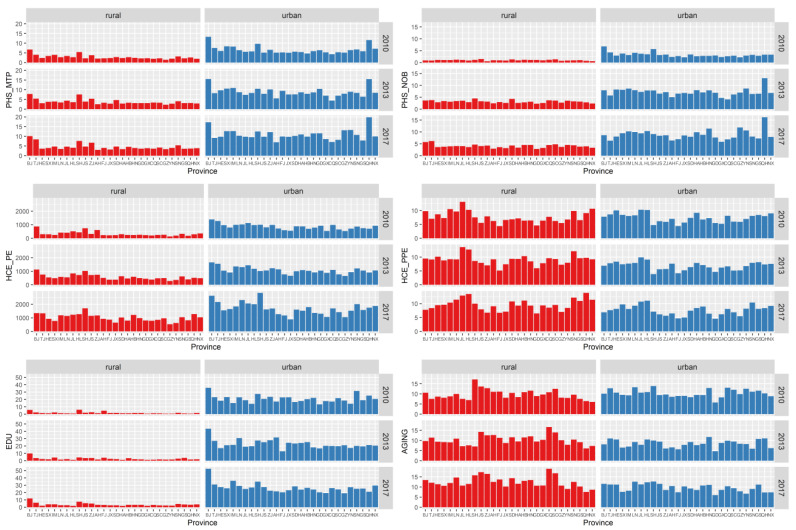
The comparison of urban and rural social environments.

**Figure 3 ijerph-19-10625-f003:**
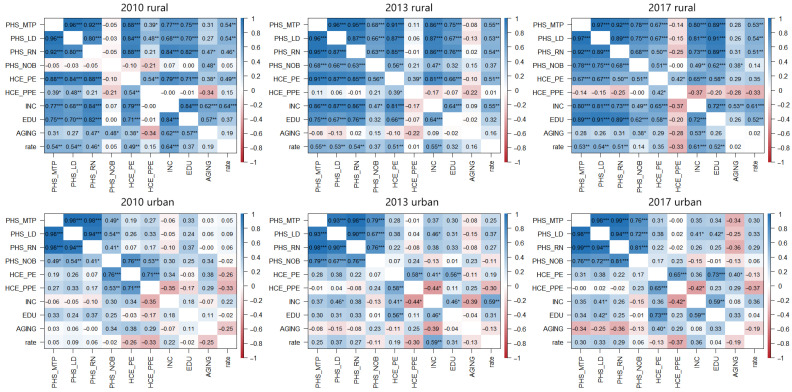
The correlation matrix among SRHS and different influence factors at the provincial level. * *p* < 0.05, ** *p* < 0.01, *** *p* < 0.001.

**Figure 4 ijerph-19-10625-f004:**
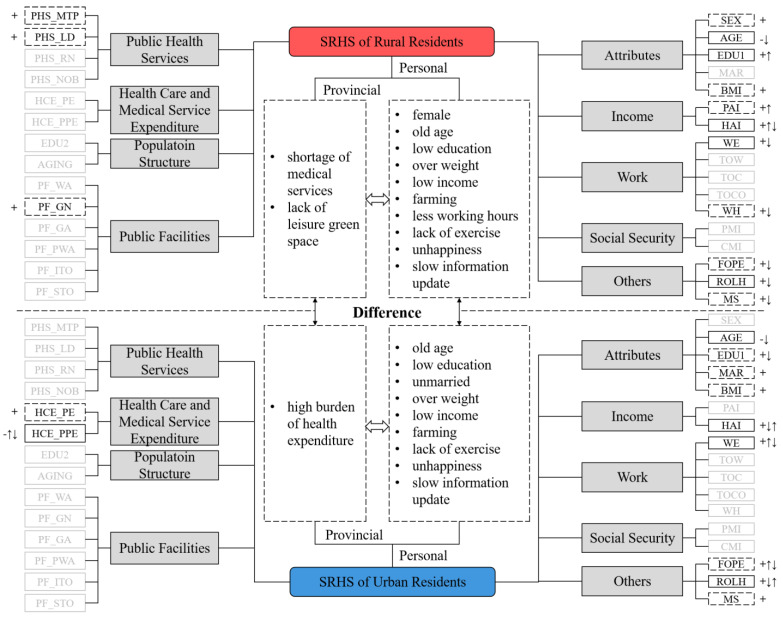
The comparison of the impact mechanism of the self-rated health status of urban and rural residents. Different types of box styles represent different influences: a solid line box with a significant impact all time; a dash line box with a significant impact in some years; an almost invisible box with no impact. The symbol “+” indicated a positive impact, and the symbol “-” indicated a negative impact. The symbol “↑” indicated the influence was gradually enhanced, and the symbol “↓” indicated the influence was gradually weakened.

**Table 1 ijerph-19-10625-t001:** Individual data.

Class	Index	Abbreviation	Item/Description
healthy	self-rated health status	SRHS	healthy/unhealthy (1/0)
location	province	-	28 provinces
	region	-	urban/rural
attribute	sex	SEX	male/female (1/0)
	age	AGE	
	education	EDU1	semi-illiterate/primary school/junior middle school/high school/college/undergraduate/graduate and above (1/2/3/4/5/6/7)
	marriage	MAR	with a lover/without a lover (1/0)
	height	HIT	the unit is the centimeter
	weight	WIT	the unit is the kilogram
	BMI	BMI	1, WIT/(HIT/100)^2 ≤ 24; 0, WIT/(HIT/100)^2 > 24
income	personal annual income	PAI	the unit is CNY
	household annual income	HAI	the unit is CNY
employment	work experience	WE	non-farm work/never work or farm work (1/0)
	type of work	TOW	full-time/part-time (1/0)
	type of company	TOC	government/non-government (1/0)
	type of company ownership	TOCO	public ownership/private ownership (1/0)
	work hours	WH	the unit is the hour
social security	public medical insurance	PMI	participate/did not participate (1/0)
	commercial medical insurance	CMI	participate/did not participate (1/0)
physical training	frequency of physical exercise	FOPE	never/several times a year/several times a month/several times a week/every day (1/2/3/4/5)
happiness	recognition of life happiness	ROLH	happiness/unhappiness (1/0)
information sources	the main source of information	MS	internet media/traditional media (1/0)

**Table 2 ijerph-19-10625-t002:** Provincial data.

Class	Index	Abbreviation	Unit
public health services	medical technical personnel per 1000 persons	PHS_MTP	person
	licensed (assistant) doctors per 1000 persons	PHS_LD	person
	registered nurses per 1000 persons	PHS_RN	person
	number of beds in health care institutions per 1000 persons	PHS_NOB	unit
health care and medical service expenditure	per capita consumption expenditure of health care and medical services	HCE_PE	CNY
	proportion of per capita health care and medical services expenditure in consumer expenditure	HCE_PPE	%
population structure	the proportion of the population with a college degree or above	EDU2	%
	the proportion of the population aged 65 and above	AGING	%
public facilities	the coverage rate of the population with access to tap water	PF_WA	%
	per capita public green areas	PF_GN	m^2^
	domestic garbage harmless treatment rate	PF_GA	%
	cumulative proportion of tap water beneficiaries	PF_PWA	%
	the proportion of national investment in rural toilet improvement in total investment	PF_ITO	%
	sanitary toilet penetration	PF_STO	%

**Table 3 ijerph-19-10625-t003:** The overall change of SRHS (%).

Year	Total			Rural			Urban		
	Number of Samples	Std	Healthy (%)	Number of Samples	Std	Healthy (%)	Number of Samples	Std	Healthy (%)
2010	8510	0.39	81.36	1586	0.43	75.22	6924	0.35	85.61
2013	8674	0.36	84.30	1362	0.40	80.22	7312	0.32	88.63
2017	10710	0.40	79.44	2202	0.46	68.96	8508	0.35	85.51

**Table 4 ijerph-19-10625-t004:** Cross analysis of health status and personal attributes considering urban and rural differences.

Index	Region	Item	2010		2013		2017	
			Unhealthy	Healthy	Unhealthy	Healthy	Unhealthy	Healthy
SEX	rural	female	30.17	69.83	22.05	77.95	33.92	66.08
		male	19.44	80.56	17.61	82.39	28.12	71.88
	urban	female	15.36	84.64	12.49	87.51	16.35	83.65
		male	13.37	86.63	10.31	89.69	12.41	87.59
BMI	rural	>24	19.93	80.07	17.59	82.41	30.54	69.46
		≤24	26.32	73.68	20.61	79.39	31.28	68.72
	urban	>24	16.36	83.64	12.92	87.08	17.40	82.60
		≤24	13.27	86.73	10.53	89.47	12.87	87.13
MAR	rural	unmarried	31.45	68.55	22.51	77.49	35.83	64.17
		married	23.76	76.24	19.31	80.69	30.04	69.96
	urban	unmarried	16.27	83.73	15.81	84.19	15.76	84.24
		married	13.94	86.06	10.16	89.84	14.10	85.90
WE	rural	farm work	28.40	71.60	25.30	74.70	36.08	63.92
		non-farm work	5.61	94.39	6.48	93.52	9.92	90.08
	urban	farm work	23.63	76.37	20.22	79.78	23.26	76.74
		non-farm work	5.91	94.09	4.77	95.23	5.54	94.46
TOW	rural	part-time	27.76	72.24	24.29	75.71	35.12	64.88
		full-time	5.22	94.78	6.13	93.87	9.22	90.78
	urban	part-time	22.80	77.20	19.42	80.58	22.10	77.90
		full-time	5.36	94.64	4.46	95.54	5.28	94.72
TOC	rural	non-government	25.10	74.90	20.20	79.80	31.50	68.50
		government	7.69	92.31	7.04	92.96	5.71	94.29
	urban	non-government	15.66	84.34	12.25	87.75	15.36	84.64
		government	5.94	94.06	4.44	95.56	6.10	93.90
TOCO	rural	private ownership	25.42	74.58	20.54	79.46	31.70	68.30
		public ownership	6.09	93.91	5.68	94.32	6.00	94.00
	urban	private ownership	16.50	83.50	13.20	86.80	15.81	84.19
		public ownership	5.77	94.23	4.12	95.88	5.66	94.34
PMI	rural	non-participate	23.74	76.26	16.82	83.18	31.97	68.03
		participate	24.87	75.13	20.01	79.99	30.97	69.03
	urban	non-participate	13.13	86.87	11.95	88.05	15.74	84.26
		participate	14.59	85.41	11.29	88.71	14.38	85.62
ROLH	rural	unhappiness	46.62	53.38	40.90	59.10	53.86	46.14
		happiness	21.83	78.17	17.70	82.30	28.16	71.84
	urban	unhappiness	33.85	66.15	27.61	72.39	36.88	63.13
		happiness	12.78	87.22	10.00	90.00	12.79	87.21
MS	rural	traditional media	25.57	74.43	21.74	78.26	37.29	62.71
		internet media	0.89	99.11	3.70	96.30	10.17	89.83
	urban	traditional media	16.90	83.10	14.75	85.25	23.42	76.58
		internet media	3.99	96.01	3.49	96.51	6.03	93.97

**Table 5 ijerph-19-10625-t005:** The results of the HLM analysis.

Level	Variable	Rural			Urban		
		2010	2013	2017	2010	2013	2017
Individual	(Intercept)	0.870	0.298	−0.493	2.407 ***	4.048 ***	1.497 ***
	SEX	0.648 ***					
	AGE	−0.048 ***	−0.032 ***	−0.032 ***	−0.038 ***	−0.038 ***	−0.032 ***
	EDU	0.151 **	0.145 **	0.216 ***	0.165 ***		0.153 ***
	MAR					0.314 *	
	BMI			0.237 **			0.305 ***
	scale (PAI)	0.243 *		0.490 ***			
	scale (HAI)	0.250 ***	0.455 ***	0.378 ***	0.241 **	0.364 ***	
	WE	0.807 ***	0.447 **	0.400 **	0.797 ***	0.874 ***	0.843 ***
	WH		0.011 ***	0.001 *			
	FOPE		0.133 **	0.112 ***	0.168 ***	0.208 ***	0.170 ***
	ROLH	1.152 ***	1.069 ***	1.032 ***	1.160 ***	1.057 ***	1.302 ***
	MS		0.552 *	0.425 **			0.293 **
Provincial	PHS_MTP	0.429 *					
	PHS_LD		0.629 *				
	scale (HCE_PE)					0.588 ***	
	HCE_PPE				−0.118 *	−0.268 **	−0.095 *
	PF_GN			0.122 ***			
Statistical Test	Marginal *R*^2^	0.327	0.321	0.382	0.298	0.341	0.311
	Conditional *R*^2^	0.359	0.378	0.399	0.320	0.389	0.331
	AIC	3246.525	3688.404	4054.724	3441.388	2415.210	4616.306
	BIC	3308.070	3758.858	4136.308	3500.099	2478.650	4684.528

* *p* < 0.05, ** *p* < 0.01, *** *p* < 0.001. “scale” means that the data had been normalized.

## Data Availability

The data of the Chinese General Social Survey (CGSS) and the Chinese Statistical Yearbook are fully open to everyone.

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
