# Peer review of "Multi-Dimensional Comparison of the Impact Mechanism of the Self-Rated Health Status of Urban and Rural Residents in Chinese Social Environments"

_ijerph, 2022, doi:10.3390/ijerph191710625_

Round 1
Reviewer 1 Report
In the study, Authors focused on crucial issue of multi-dimensional comparison of the impact mechanism of the self-rated health status of urban and rural residents under different social environments in China. the paper systematically analyzed and compared the impact mechanism of the SRHS of urban and rural residents from multiple dimensions such as time, space, and scale, based on the data of the Chinese General Social Survey (CGSS) and China Statistical Yearbook. Remarks: Are there concrete steps that can be recommended for the authorities? The advantages and novelty of the research approach need to add. This will help highlight any unique findings. What about the years after 2017? On what base the attributes were chosen (presented in the table 1)?
Author Response
Reviewer 1
- Are there concrete steps that can be recommended for the authorities?
Response: We have added some recommendations for improving the health of rural residents and areas.
“Therefore, in response to the problems of happiness, green space, and the widespread ag-ing of rural areas that rural residents care about, improvement measures are needed in the following areas. First, we should carry forward traditional culture and make full use of the rural stage. While enhancing the sense of local identity and belonging, this would also support the health of the villagers. Second, with rural ‘left-behind’ women as the main labor force, a new mode of home-based care for the elderly should be explored in rural areas under the unified arrangement and supervision of village committees. The third is to deepen the hierarchical diagnosis and treatment system. This would ensure that common and chronic diseases are treated conveniently and efficiently in the county, ultimately improving rural residents’ willingness to seek medical treatment.”
- The advantages and novelty of the research approach need to add. This will help highlight any unique findings.
Response: We have further emphasized the necessity to integrate time, space, and scale in the study of the SRHS to show the advantages and novelty of the research: “Considering time and space—but lacking scale—makes it hard to explore the relationship between individual health and living environment. When time and scale are considered—but not space—the differences in health status between regions cannot be explained. Considering space and scale only, it is impossible to conclude the changing trend of influencing factors. SRHS is a result of the comprehensive effect of people’s perception of the social environment, and study from a single dimension may overemphasize the influence of a certain factor.”
- What about the years after 2017?
Response: Since the latest data from CGSS is up to 2017, this article does not cover the situation after 2017. But some new information about health and medical care is added in the revised article.
“The cost of medical insurance in China has nearly doubled from 180 CNY in 2017 to 350 CNY in 2022—for all residents. For rural residents with an annual per capita income of less than 20,000 CNY, that burden is very heavy.”
- On what base the attributes were chosen (presented in the table 1)?
Response: In the introduction, we have indicated that income, education, and job are the main factors, and also summarized the directions of in-depth research which included living conditions, health literacy, medical insurance, green space, and other factors. Except that, for some factors chosen in this article which are different from previous research, such as happiness and information sources, we have already explained the reasons and purposes. And all of them have been cited with references.

Reviewer 2 Report
This is an interesting paper which compared the impact mechanism of the self-rated health status (SRHS) of urban and rural residents from multiple dimensions such as time, space, and scale, based on the data of the Chinese General Social Survey (CGSS) and China Statistical Yearbook. However, the document needs to be re-structured a bit so it can be more clear contribution to current knowledge. In particular, it is needed to improve several methodological aspects, clarify the results section, and focus the discussion according to the objectives of the study. Moderate English changes are also required. Few references are mentioned in this paper, while many references should be listed.
Other detailed comments are below, I hope, will be helpful as you work on this paper.:
Background:
1. Page 1, line 29-31, “Although urbanization has brought social and economic prosperity and improved medical and health conditions, it has also widened the omnidirectional gap between urban and rural development. ” - Reference please.
2. Page 3, line 59, it would be helpful for the readers if there was a brief explanation and definition about “work” later.
3. Page 2, line 59-94, it is necessary to highlight the influences of main factors and describe clearly.
Method:
1. Page 3-6, The authors described the sample, however, what about the exclusion? how did the authors deal with the missing data of the variables? It is necessary to clarify in the methods.
2. Page 3, line 116-117, “The data used in this paper come from the Chinese General Social Survey (CGSS) in 2010, 2013, and 2017.” and Page 3, line 147-148, “After removing some questionnaires with incomplete information, 46,606 sample data were finally sorted out”. Did the authors use the weight data and how?
3. Please clarify the statistical software and its manufactured, city and country from where software has been source.
4. Page 6, line 199, “it is necessary to perform ‘Logit Transformation’ on the dependent variable.” - Reference please.
5. Did the authors consider the interaction between variables?
Results:
1. Page 6, line 202, the descriptive statistics should be presented at beginning. Please list the numbers of each wave, also in table 3.
2. The authors used the Chinese General Social Survey (CGSS) in 2010, 2013, and 2017. However, the changes over years were not described.
3. The authors performed HLM analysis while it was not introduced in the methods part.
Discussion & Conclusion:
1. Page 12, line 342-358, “Regional differences cannot be ignored.” It is obviously to know the differences from different levels, which kind of regional difference should be described clearly. The authors also need to point out inequalities among the rural and urban areas.
2. Many references should be listed in discussion, however, the authors did not.
3. Page 15, line 400-401, “health is not only an individual-level matter but also closely related to the regional social environment.” But there is no strong evidence or explanation to support the conclusion from the authors’ study in this paper. The authors also need to analysis and compare the regional social environment in results.
Author Response
Reviewer 2
Editorial Certificate
The manuscript below was edited for correct English language usage, grammar, punctuation and spelling by qualified native English speaking editors at Charlesworth Author Services. Please see the editorial certificate in the word file.
Background:
- Page 1, line 29-31, “Although urbanization has brought social and economic prosperity and improved medical and health conditions, it has also widened the omnidirectional gap between urban and rural development. “- Reference please.
Response: We have added a reference “Chen, M.; Zhou, Y.; Huang, X.; Ye, C. The Integration of New-Type Urbanization and Rural Revitalization Strategies in China: Origin, Reality and Future Trends. Land 2021, 10, 207, doi:10.3390/land10020207.” And this article mentioned: “It is believed that urban-biased urbanization has widened the development gap between urban and rural areas since reform and opening up.”
- Page 3, line 59, it would be helpful for the readers if there was a brief explanation and definition about “work” later.
Response: To make the definition accurate and clear, we have replaced the word “work” with “employment”, and also explained the main factors. “For example, income includes many types such as absolute and relative, personal and family. … Education usually refers to the highest educational level achieved. … Employment focuses on the employer-employee relationship and job characteristics like work experience and hours.”
- Page 2, line 59-94, it is necessary to highlight the influences of main factors and describe clearly.
Response: In the in-depth discussion on the health influencing factors by scholars, this article has pointed out two main directions. “Scholars have approached this complexity through in-depth research in different fields, which can be roughly divided into two areas. One area focuses on individuals and families, examining factors like their living conditions, health literacy, and medical insurance. The main conclusions from this approach often stand in contrast to common sense.”, and next we have explained the “anti-common sense” for living conditions, health literacy, and medical insurance. For the second direction, we have added some descriptions in the revision: “Now, for example, links between green space and health are widely acknowledged[18].” Similarly, we also have added some descriptions for urban and rural disparities in China: “In terms of urban and rural disparities in China, studies have shown that urban and rural residents exhibit non-linear differences in SRHS and that SRHS has a slight urban advantage[23].”
Method:
- Page 3-6, The authors described the sample, however, what about the exclusion? how did the authors deal with the missing data of the variables? It is necessary to clarify in the methods.
Response: For individual data, “Where key information in the questionnaire was missing, these data were deleted. After removing incomplete data, a total of 27,894 sample data remained. Further inspection of data distribution showed that the distribution of sample data across provinces was consistent for the different years, with a mean percentage of 3.57% for each province in each year. Therefore, the missing values had no significant effect on the study.” There was a mistake in the number of samples, the correct number is 27894 (including the years 2010, 2013, and 2017) not 46606 (including the years 2010, 2012, 2013, 2015, and 2017). In the first manuscript, we forgot to exclude the number of samples in two years. This is only a clerical error and will not affect the subsequent analysis. For provincial data, “Changes in statistical quality meant that alternative indicators were selected for some years. Among them, PF_WA, PF_GN, and PF_GA were used for urban data each year but only for rural data in 2017. Instead, PF_PWA, PF_ITO, and PF_STO were used in 2010 and 2013 for rural data.”
- Page 3, line 116-117, “The data used in this paper come from the Chinese General Social Survey (CGSS) in 2010, 2013, and 2017.” and Page 3, line 147-148, “After removing some questionnaires with incomplete information, 46,606 sample data were finally sorted out”. Did the authors use the weight data and how?
Response: If the “weight” refers to the statistical weight, we did not assign any additional weight to the raw data of CGSS. And if the “weight” refers to the body weight, we have used BMI in the later analysis, which is calculated from height and weight data.
- Please clarify the statistical software and its manufactured, city and country from where software has been source.
Response: “Data analysis took place using R, a free software environment for statistical computing and graphics, with a range of packages for researchers. In this research, we used the package ‘bruceR’ (version 0.8.5) for HLM analysis based on R (version 4.1.3).”
- Page 6, line 199, “it is necessary to perform ‘Logit Transformation’ on the dependent variable.” - Reference please.
Response: We have added a reference “Wang, J. Hours Underemployment and Employee Turnover: The Moderating Role of Human Resource Practices. null 2018, 29, 1565–1587, doi:10.1080/09585192.2016.1203346.” And this article indicated that: “Logit analysis was used because the dependent variable (turnover) contains dichotomous values (0 or 1).”
- Did the authors consider the interaction between variables?
Response: Actually, we have considered the interaction between variables, but the results were not ideal and interaction analysis is not suitable for this article. First, the interaction effect between individual and provincial levels was very weak, because of the huge range and scale of a province in China. Second, the interaction effect was hardly explained in a different year and there was no obvious trend to conclude the rule. Third, the cross analysis can also reflect the interaction to a certain extent.
Results:
- Page 6, line 202, the descriptive statistics should be presented at beginning. Please list the numbers of each wave, also in table 3.
Response: Because the SRHS is a binary variable, the percentage of the value “1” is equal to the mean of the sequence, we just added the number of samples and standard deviation for the descriptive statistics in table 3, as well as some descriptions in the text: “From 2010 to 2017, the sample quantity increased, but the structure remained the same. In 2013, the ratio of rural to urban data was 1.5:8.5, and for the other 2 years the ratio was 2:8 (rural to urban). The change in standard deviation shows that the SRHS difference among Chinese residents was expanding, especially for rural residents whose annual standard deviation was larger than the full sample level. The proportion of healthy groups of Chinese residents declined but remained at a relatively high level (Table 3). The health status of urban residents was relatively stable while the proportion of unhealthy rural residents had increased significantly, providing the main reason for the decline of the overall health level.”
- The authors used the Chinese General Social Survey (CGSS) in 2010, 2013, and 2017. However, the changes over years were not described.
Response: Time trend is an important study perspective in our article, so except for table 1 and table 2 which only show the index system, other tables and figures all include time trend. However, the changes over years in spatial analysis and HLM analysis were described more clearly than that in cross analysis. So we have improved the descriptions in cross analysis:
- “As a result, the health rate of unmarried women in rural areas was below 70%, dropping to 66.08% in 2017.”
- “Differences in health status caused by occupational attributes were the most prominent. More than 90% of residents engaged in non-farm work in urban or rural areas were consistently healthy.”
- “The SRHS of people who used new media as the main source of information reached an excellent level in the initial stages (2010 and 2013), but it declined significantly in 2017.”
- The authors performed HLM analysis while it was not introduced in the methods part.
Response: We have already described the characteristics of HLM analysis in the method part: “However, homogeneity of variance and independent distribution do not hold for multi-level nested data. Thus, we used a hierarchical linear model (HLM), widely used in environmental and health research[33].” And we also added a description with a reference “Williams, B.L.; Pennock-Roman, M.; Suen, H.K.; Magsumbol, M.S.; Ozdenerol, E. Assessing the Impact of the Local Environment on Birth Outcomes: A Case for HLM. J. Expo. Sci. Environ. Epidemiol. 2007, 17, 445–457, doi:10.1038/sj.jes.7500537.” This article indicated that: “Hierarchical linear Models (HLM) is a useful way to analyze the relationships between community level environmental data, individual risk factors, and birth outcomes. With HLM we can determine the effects of potentially remediable environmental conditions (e.g., air pollution) after controlling for individual characteristics such as health factors and socioeconomic factors.”
Discussion & Conclusion:
- Page 12, line 342-358, “Regional differences cannot be ignored.” It is obviously to know the differences from different levels, which kind of regional difference should be described clearly. The authors also need to point out inequalities among the rural and urban areas.
Response: We have rewritten the first part of the discussion.
“(1) Narrowing the gap between urban and rural areas is key to improving regional health. The vast gap between urban and rural areas—whether at the individual or provincial level—and the overall shortcomings of rural areas significantly affect the health level of China and its regions. More concerning, the gap between urban and rural areas has not narrowed significantly[35] and the health of rural residents continues to deteriorate. In the process of improving the health of rural residents by increasing their income, a population migration route from rural areas in undeveloped provinces to cities in undeveloped provinces to cities in developed provinces has gradually formed, making urban-rural differences deeply bound to provincial differences[36]. The fact that the spatial pattern of health rate was consistent with the pattern of China’s social and economic development illustrates this point. The resulting cumulative effect has caused an imbalance in the regional high-quality labor force and also enhanced people’s sensitivity to the social environment, magnifying the influence of the social environment on health in regional comparisons.”
- Many references should be listed in discussion, however, the authors did not.
Response: Not only the discussion, we have added 15 references to the revised manuscript, and the total number of references reaches 41. And details can be found in the revised manuscript, which will not be listed here.
- Page 15, line 400-401, “health is not only an individual-level matter but also closely related to the regional social environment.” But there is no strong evidence or explanation to support the conclusion from the authors’ study in this paper. The authors also need to analysis and compare the regional social environment in results.
Response: According to the revised discussion, we have modified the expression of some conclusions. The sentence mentioned here has been modified: “Multi-dimensional comparative analysis provided a more comprehensive and in-depth understanding of the impact mechanism of SRHS, and strengthened understanding that narrowing the gap between urban and rural areas in China is key to improving regional health.”

Round 2
Reviewer 2 Report
As requested in the guidelines for revision, the authors have tried to amend this article in detailed accordingly. Necessary references have been added. However, the authors still need to take actions to meet the required English language competency level for publication.
Good luck.